DATA RELEASE

# A reference assembly for the legume cover crop hairy vetch (*Vicia villosa*)

Tyson Fuller[1,†], Derek M. Bickhart[1,†], Lisa M. Koch[1], Lisa Kissing Kucek[1], Shahjahan Ali[1], Haley Mangelson[2], Maria J. Monteros[3], Timothy Hernandez[3], Timothy P. L. Smith[4], Heathcliffe Riday[1] and Michael L. Sullivan[1,*]

1  US Dairy Forage Research Center, United States Department of Agriculture Agricultural Research Service (USDA-ARS), 1925 Linden Drive, Madison, WI 53706, USA
2  Phase Genomics, 1617 8th Ave N, Seattle, WA 98109, USA
3  Noble Research Institute, 2510 Sam Noble Parkway, Ardmore, OK 73401, USA
4  US Meat Animal Research Center, United States Department of Agriculture Agricultural Research Service (USDA-ARS), PO Box 166 (State Spur 18D), Clay Center, NE 68933, USA

## ABSTRACT

*Vicia villosa* is an incompletely domesticated annual legume of the Fabaceae family native to Europe and Western Asia. *V. villosa* is widely used as a cover crop and forage due to its ability to withstand harsh winters. Here, we generated a reference-quality genome assembly (Vvill1.0) from low error-rate long-sequence reads to improve the genetic-based trait selection of this species. Our Vvill1.0 assembly includes seven scaffolds corresponding to the seven estimated linkage groups and comprising approximately 68% of the total genome size of 2.03 Gbp. This assembly is expected to be a useful resource for genetically improving this emerging cover crop species and provide useful insights into legume genomics and plant genome evolution.

**Subjects**  Genetics and Genomics, Bioinformatics, Plant Genetics

**Submitted:**  29 March 2023

*  Corresponding author. E-mail: michael.sullivan@usda.gov

†  Contributed equally.

Preprint submitted at https://doi.org/10.1101/2023.03.28.534423

# DATA DESCRIPTION

## Background

*Vicia villosa* Roth (hairy vetch) is a mostly outcrossing hermaphroditic diploid ($2n = 2x = 14$) annual legume originating from Europe and Western Asia [1, 2]. *V. villosa* belongs to the *Vicia* genus of the Fabaceae family and is the second most cultivated vetch species worldwide, with value both as a forage species and as a cover crop [1, 3, 4]. *V. villosa* is especially useful as a winter cover crop for warm season crops (i.e., corn [5] and soybeans [6]) since it is one of the few legumes that can survive in harsh winter conditions [7].

*V. villosa's* use as a cover crop benefits cash crops primarily through nitrogen fixation, soil and water conservation, and its ability to produce biomass in a short period [3, 4, 7]. *V. villosa* is an incompletely domesticated species. Variations in pod dehiscence and seed dormancy across populations can result in reduced yields and increased weediness [8, 9], which limits the adoption of *V. villosa* use by farmers [8, 10].

Differences in chromosome number between species of the *Vicia* genus have been identified, making it an interesting model for studies of the plant genome [2, 11, 12].

Reference genomes for species within the *Vicia* genus can be used to better understand the phylogeny and karyotype evolution of different species within the genus. Species-specific reference genomes can also inform the identification of genes involved in beneficial and undesirable traits, ultimately increasing their use as cover crops by farmers. However, the first chromosome-level genome assembly within the *Vicia* genus (*Vicia sativa*, or common vetch) has only recently been published [13].

The high heterozygosity of *V. villosa*, presumably due to its outcrossing nature, presents a unique challenge to generate high-quality genome assemblies with current assembly methods. Heterozygous regions result in both false duplications of sequences and less contiguous assemblies [14–17]. This adversely impacts the final assembly size and other downstream analyses, such as gene prediction and functional annotation [14, 17]. We circumvent these difficulties by applying low error-rate long-read sequencing along with both manual and automated curation. This method allowed us to generate a high-quality reference genome for the highly heterozygous *V. villosa*.

## Context

We present a high-quality reference genome assembly for *V. villosa*, which is only the third reference-quality genome assembly in the *Vicia* genus after those of *V. sativa* [13] and *Vicia faba* L. [18]. Our assembly was compared with those of other legume species, including *V. sativa*. We observed a markedly higher level of heterozygosity in *V. villosa* compared to *V. sativa*, a self-crossing member of the *Vicia* genus. We demonstrated that the *V. sativa* reference is unsuitable as a proxy for variant calling with the DNA sequence data of *V. villosa* despite their common lineage. Our assembly, Vvill1.0 represents a reference-quality genomics resource for this common cover crop species, and provides further evolutionary insights into a unique clade of leguminous plant species.

## METHODS

### Sample information, nucleic acid extraction, and library preparation

A single individual was chosen from the 'AU Merit' [19] cultivar for its ability to be clonally propagated in tissue culture and was named 'HV-30'. This individual of *V. villosa* was used for long-read and short-read DNA sequencing (Figure 1). Approximately 0.75 g of frozen leaf tissue from an individual plant was ground with mortar and pestle under liquid nitrogen. High-molecular-weight DNA was extracted using the NucleoBond HMW DNA extraction kit as directed by the manufacturer (Macherey Nagel, Allentown, PA, USA). The DNA pellet was resuspended in 150 µL of 5 mM Tris-Cl pH 8.5 (kit buffer HE) by standing at 4 °C overnight, with integrity estimated by fluorescence measurement (Qubit, Thermo Fisher, Waltham, MA, USA), optical absorption spectra (DS-11, DeNovix, Willmington, DE, USA), and size profile (Fragment Analyzer, Thermo Fisher).

High molecular weight DNA, used for high-fidelity long-read sequencing on the Pacific Biosciences (Menlo Park, CA, USA) Sequel II platform (HiFi sequence), was sheared (Hydroshear, Diagenode, Denville, NJ, USA) using a speed code setting of 13 to achieve a size distribution with "peak" at approximately 23 kbp. Smaller fragments were removed by size selection for >12 kbp fragments (BluePippin, Sage Science, Beverly, MA, USA). Size-selected DNA was used to prepare four SMRTbell libraries using the SMRTbell Express Template Prep Kit 2.0, as recommended by the manufacturer (Pacific Biosciences).

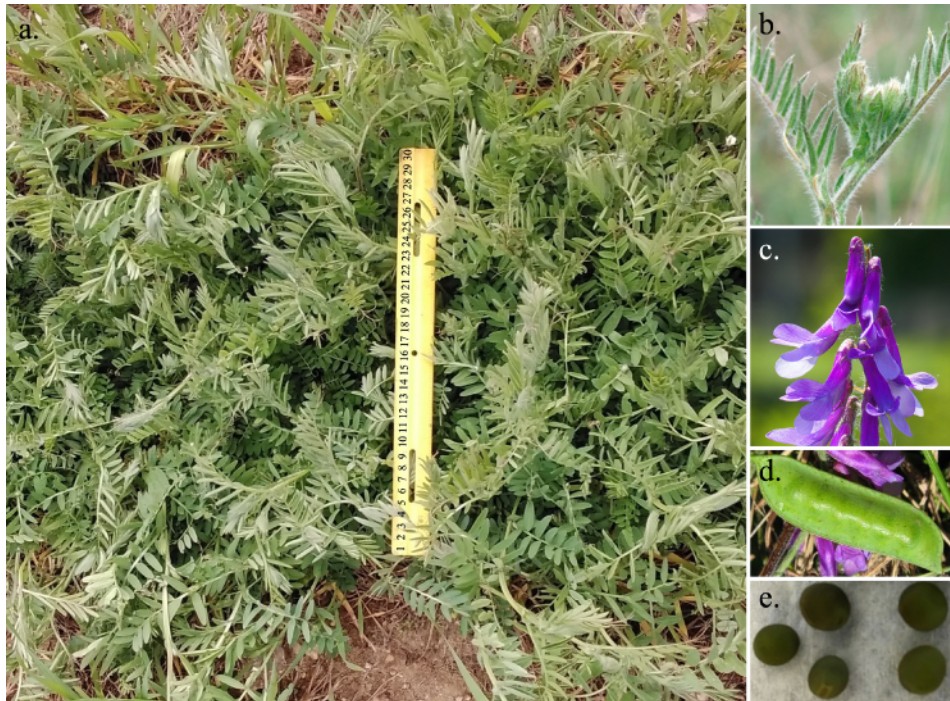

**Figure 1.** The HV-30 genotype of *Vicia villosa* was selected from the cultivar 'AU Merit' [19]. Panel (a) shows 'AU Merit' growing in Beltsville, Maryland on March 30th, 2022. A yellow 30 cm ruler is in the middle of the image for scale. The photo was taken by Allen Burke of USDA-ARS Beltsville Agricultural Research Center. Panels (b), (c), (d), and (e) show leaves, flowers, pod, and seeds of 'AU Merit', respectively.

The DNA for short-read sequencing was sheared to 550 bp on a Covaris M220 focused-ultrasonicator (Covaris, Woburn, MA, USA) by the University of Wisconsin-Madison Biotechnology Center (Madison, WI, USA), as specified in the TruSeq DNA PCR-Free Reference Guide (Illumina, San Diego, CA, USA) [20]. A library was prepared using 2 μg of the sheared DNA with the TruSeq DNA PCR-Free Library Preparation Kit, according to the manufacturer's guidance.

## Genome assembly and scaffolding

A list of the software tools and versions used in this analysis is provided in Table 1. Genomic short-read libraries were sequenced on a NextSeq 500 instrument (Illumina) with a NextSeq High Output v2 300 Cycle Kit, generating 982 million 2× 150 paired-end (PE) reads. This resulted in 147.81 Gbp of genomic sequences. These reads were used to estimate the total assembly length and heterozygosity of the sequenced *V. villosa* genotype. An abundance histogram of 21-base length k-mers derived from the reads was generated from *V. villosa* short-read data using the Jellyfish version 2.2.9 tool [21]. The histogram was then uploaded to the GenomeScope tool (RRID:SCR_017014) [22, 23], which estimated the haploid genome size to be 1,629 Mbp when using over 1,000,000 max k-mer count entries in the model. The expected genome size of *V. villosa* (2.0 Gbp) [24] is much larger, but k-mer-based estimations are generally underestimations. A recent survey of the genome size in the Coleoptera revealed a similar genome size underestimation by k-mer modeling compared to flow-cytometry estimates [25]. The estimated heterozygosity of *V. villosa* is 3.14% (Figure 2), which is substantially higher than that reported for *V. sativa* (0.09%) [13]. High degrees of

**Table 1.** Software and versions used in assembly and analysis of Vvill1.0.

| Software | Version |
|---|---|
| BUSCO | 5.3.2 |
| BWA-MEM | 0.7.17-r1188 |
| DIAMOND | 2.0.14.152 |
| EDTA | 2.0.0 |
| EggNOGmapper | 2.1.8 |
| FRC_align | 1.0.0 |
| Freebayes | 1.3.1 |
| GenomeScope | 1.0.0 |
| Jellyfish | 2.2.9 |
| Juicebox | 2.20.00 |
| LUMPY-SV | 0.3.1 |
| Merqury | 1.3 |
| Meryl | 1.4 |
| Minimap2 | 2.24 |
| Orthofinder | 2.5.4 |
| PacBio IPA | 1.3.1 |
| PacBio SMRT Link | 9.0 |
| purge_dups | 1.0.1 |
| RepeatMasker | 4.0.6 |
| RepeatModeler (RRID:SCR_015027) | 2.0.4 |
| SAMBLASTER | 0.1.26 |
| SAMtools | 1.15.1 |
| STAR (RRID:SCR_004463) | 2.7.9 |
| UpSetR | 1.4.0 |

heterozygosity present a substantial challenge for genome assembly with higher error-rate long-reads since errors and allelic variation are indistinguishable [26]. To circumvent this issue, low-error long-reads were used as the primary vehicle for genome assembly. A total of six single-molecule real-time sequencing (SMRT) cells were used with an average insert length of 16.7 kbp. Through this method, we generated a total of 85.8 Gbp of sequence after processing for HiFi reads using the SMRT Link software version 9.0 with default settings (Pacific Biosciences). *V. villosa* primary contigs were generated using the PacBio IPA assembler (version 1.3.1, RRID:SCR_021966). Haplotigs were then screened for additional heterozygous duplications with purge_dups (version 1.0.1, RRID:SCR_021173) [27], which identified 54 Mbp of duplicated sequences [28]. All duplicated sequences were removed from the primary haplotig assembly before scaffolding, resulting in 5,373 contigs with an N50 of approximately 600 kbp (Table 2). These haplotigs represent a singular haplotype (or a mixture of haplotypes) from the sequenced individual that was resolved down to unique structural differences between sister chromatid pairs. Without a linkage map or parental single nucleotide polymorphism data, it is difficult – and likely meaningless – to ascribe a parent-of-origin to each haplotig. To assess the suitability of the assembled sequence as a reference genome for the species, we used additional datasets to create scaffolds approximating the linkage group sequences for *V. villosa*.

Assembly scaffolding consisted of a combination of automated and manual processes. Chromatin conformation capture data was generated using a Phase Genomics (Seattle, WA, USA) Proximo Hi-C 4.0 Kit, a commercially available version of the Hi-C protocol [29]. Intact cells from the sample were crosslinked using a formaldehyde solution as per the manufacturer's protocol, digested using a cocktail of restriction enzymes (DpnII, DdeI, HinfI, and MseI), end-repaired with biotinylated nucleotides, and proximity ligated to

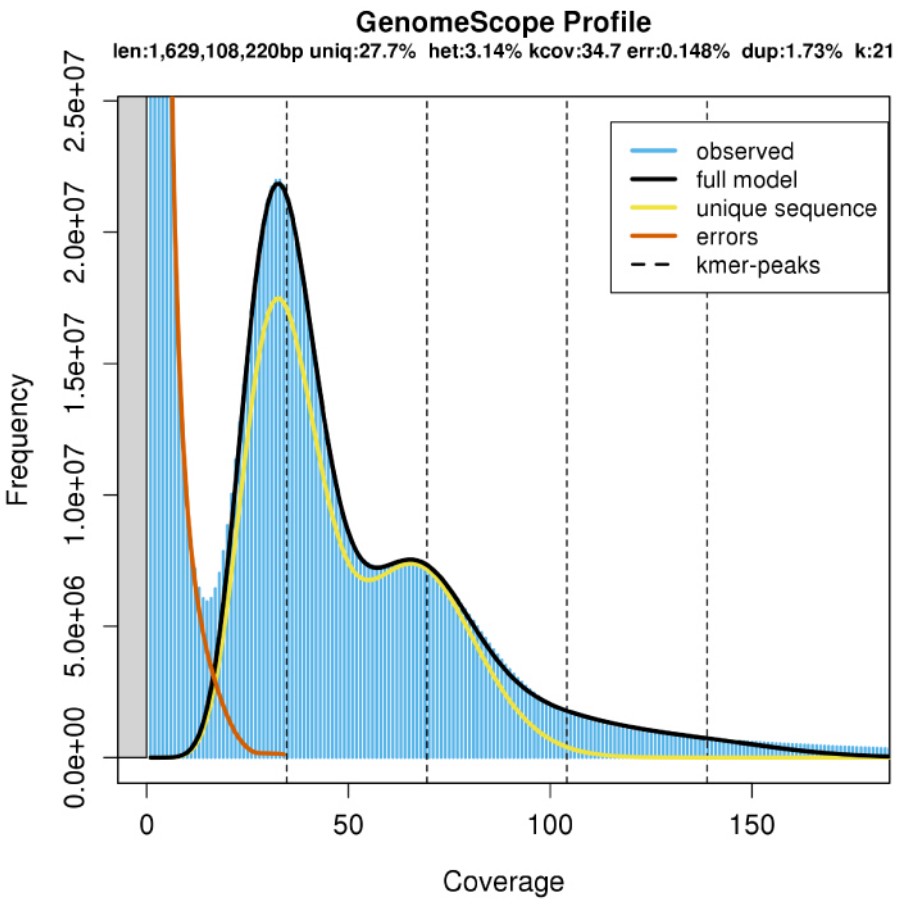

**Figure 2.** GenomeScope k-mer profile of *V. villosa* short-read data.

| Table 2. Overview of our *Vicia villosa* genome assembly. | |
|---|---|
| **Feature** | **Value** |
| Assembly size | 2,034,988,938 bp |
| No. of scaffolds | 1,888 |
| No. of contigs | 5,373 |
| Contig N50 | 604,665 bp |
| Scaffold N50 | 174,244,450 bp |
| Pseudomolecule (scaffold) size | 1,384,960,116 bp |
| Contigs anchored to pseudomolecules (number) | 3,296 |
| Contigs anchored to pseudomolecules (length) | 1,384,611,616 bp |
| GC content (%) | 35.62 |
| **Sequence data generated** | **Value (coverage)** |
| Illumina short-read WGS | 147.81 Gbp (74×) |
| Illumina short-read Hi-C | 42.14 Gbp (21×) |
| PacBio Sequel II HiFi | 85.80 Gbp (43×) |

create chimeric molecules composed of fragments from different regions of the genome that were physically proximal *in vivo*. Molecules were pulled down with streptavidin beads and processed into an Illumina-compatible sequencing library, as recommended by the

protocol. Sequencing was performed on an Illumina NovaSeq, generating 140,472,036 2× 150 PE reads.

Reads were aligned to the primary haplotig assembly following the manufacturer's recommendations [30]. Briefly, reads were aligned to the haplotig assembly using BWA-MEM (RRID:SCR_010910) [31] with the -5SP and -t 8 options specified, and all other options set to their default values. SAMBLASTER (RRID:SCR_000468) [32] was used to flag PCR duplicates, which were later excluded from analyses. Alignments were then filtered with SAMtools (RRID:SCR_002105) [33] using the -F 2304 filtering flag to remove non-primary and secondary alignments. Putative misjoined contigs were broken using Juicebox (RRID:SCR_021172) [34, 35] based on the Hi-C alignments. A total of 192 breaks were introduced, and the same alignment procedure was repeated from the beginning on the resulting corrected assembly.

A Phase Genomics' Proximo Hi-C genome scaffolding platform was used to create chromosome-scale scaffolds from the corrected assembly, as described by Bickhart *et al.* [36]. As in the LACHESIS method (RRID:SCR_017644) [37], this process computes a contact frequency matrix from the aligned Hi-C read pairs, normalized by the number of restriction sites on each contig, and constructs scaffolds in such a way as to optimize expected contact frequency and other statistical patterns in Hi-C data. Approximately 60,000 separate Proximo runs were performed to optimize the number of scaffolds and scaffold construction in order to make the scaffolds as concordant with the observed Hi-C data as possible. Juicebox was used a second time to correct scaffolding errors. Hi-C contact maps showed few off-diagonal contacts, in agreement with the final scaffold structure (Figure 3). The few off-diagonal contacts in the scaffold order are almost exclusively present on the telomeric ends of scaffolds, indicating they may be a biological signal from telomeric "bouquets" instead of scaffolding errors [38]. To our knowledge, the final scaffolded assembly Vvill1.0 is the first reference-quality genome assembly for a heterozygous out-crossing plant species in the *Vicia* genus [39].

The Vvill1.0 assembly is 2,034,988,938 bp in 1,888 scaffolds. This assembly is substantially larger than the GenomeScope haploid genome size estimate of 883 Mbp (Figure 2) but congruent with expectations from previous estimates [24]. The assembly had a scaffold N50 of 174.24 Mbp and a GC content of 35.62%; however, the contig N50 of the assembly was 604 kbp, similar to the *V. sativa* reference genome assembly (Table 2). Seven scaffolds of Vvill1.0 correspond to haploid representations of the seven estimated linkage groups of *V. villosa* [2] and comprise 67.74% of the total genome assembly size (Table 2) (Figure 4A). A substantial proportion of the assembly (~33% of all base pairs; 1,881 scaffolds) could not be placed on distinct linkage group scaffolds due to the inherent heterozygosity of the individual. Hence, a combination of orthogonal quality assessment tools for genome assembly was used to validate the completeness and accuracy of the assembly.

## DATA VALIDATION AND QUALITY CONTROL

All assembly validation and quality control data were produced by the Themis-ASM pipeline [40] run on the Vvill1.0 and *V. sativa* [13] genome assemblies with default settings. A long terminal repeat (LTR) assembly index (LAI) score was generated for Vvill1.0 using the LTR_Finder software package (RRID:SCR_015247) [41]. Vvill1.0 was predicted to have an LAI of 22.5, corresponding to the "gold" category of high-quality reference genomes based on the assembly fidelity of repeat elements [41]. A sliding window analysis of the regional

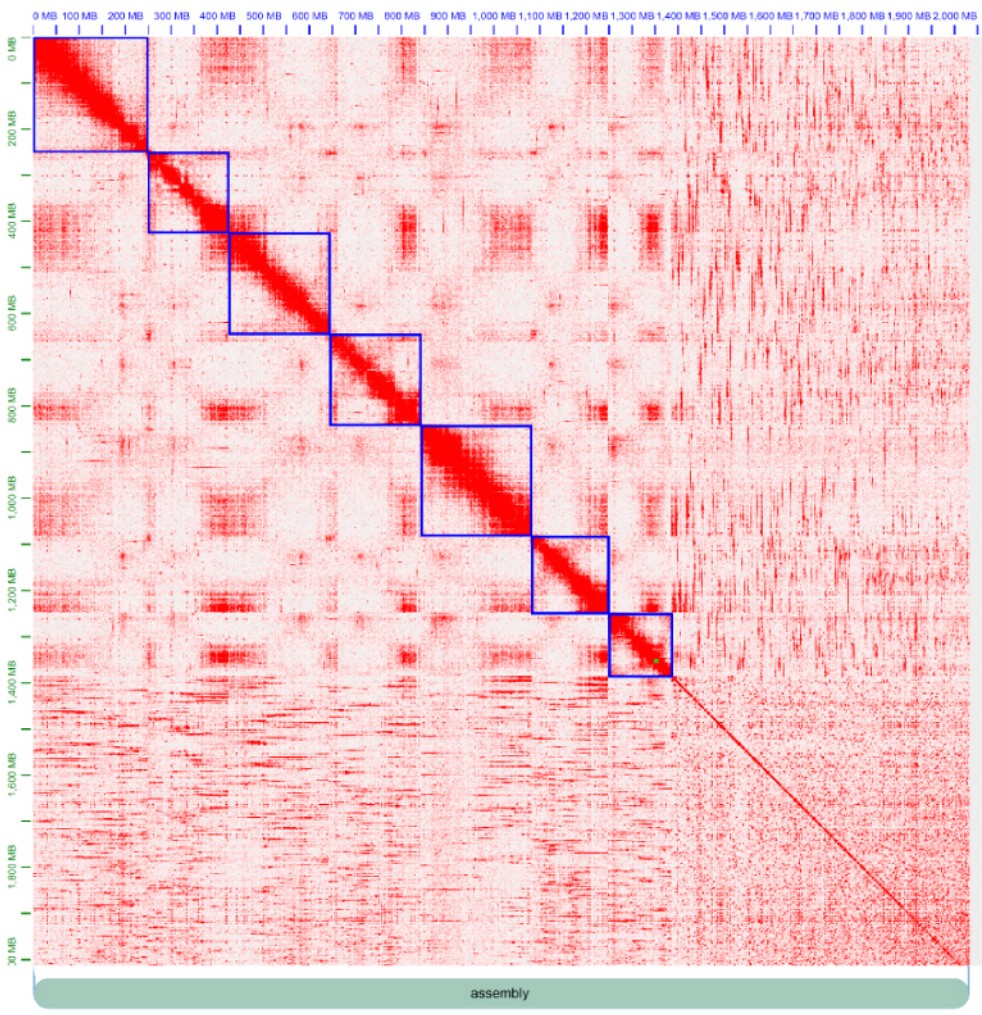

**Figure 3.** Hi-C link heatmaps and scaffold edits were produced by the JuiceBox tool [34]. Scaffold assignments (blue boxes) were identified from an optimal signal arrangement along the diagonal. Unscaffolded contigs mainly consist of very small contigs (<5 kbp), where it is less likely there will be significant Hi-C linkage data aligning to such small sequences.

LAI values on the assembly revealed only a few regions that fell below this genome-wide LAI value, possibly indicating the misassembly of repetitive regions (Figure 5). Single-copy orthologous genes were identified using the BUSCO software package (RRID:SCR_015008) [42], with the eudicots_odb10 dataset (2,326 markers) for both assemblies. Both Vvill1.0 (99% complete and duplicated BUSCOs) and *V. sativa* (98.2%) had high BUSCO completeness scores (Figure 4B); however, the Vvill1.0 assembly had a higher rate of BUSCO duplication (36.8%) than *V. sativa* (7.4%). To assess the utility of using each *Vicia* reference genome for sequence alignment for *V. villosa* resequencing studies, the *V. villosa* short-read dataset was aligned to each assembly using the BWA and SAMtools software packages [33, 43]. Short-read alignments revealed that 98.6% of the *V. villosa* reads mapped to the Vvill1.0 assembly; however, only 47.0% of the *V. villosa* reads mapped to the *V. sativa* assembly. Similar comparisons using short-reads from *V. sativa* revealed a mapping rate of 64.0% and 99.7% to the Vvill1.0 and *V. sativa* reference assemblies,



(A)

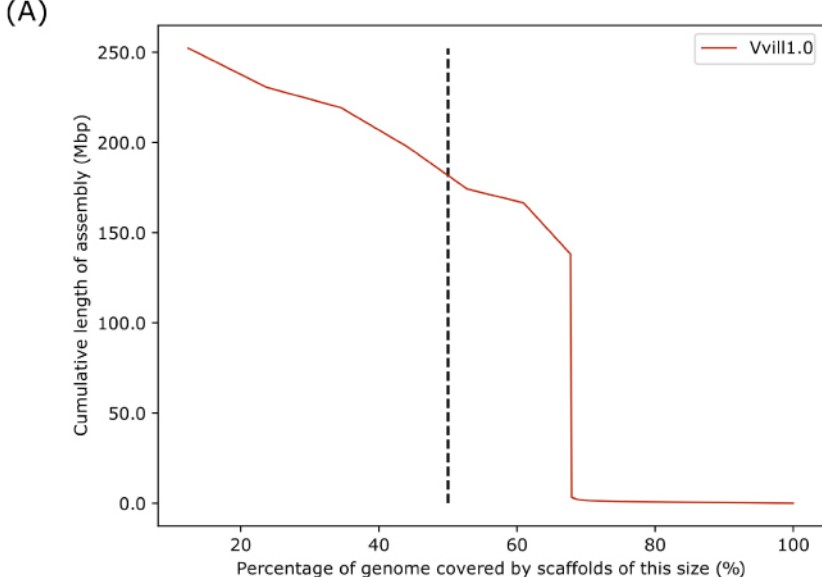

(B)

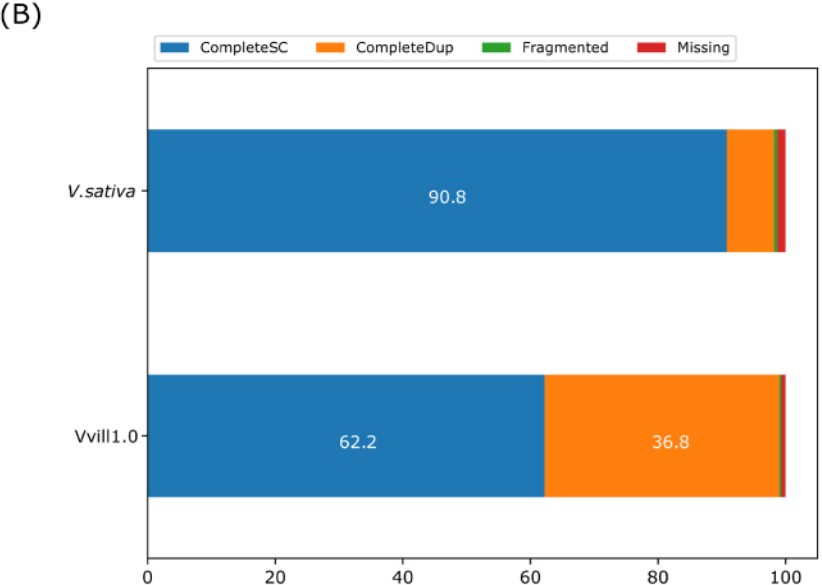

**Figure 4.** (A) Scaffold N(X) plot displaying the percentage of the genome (*x*-axis) covered by scaffolds of a specific length (*y*-axis). The vertical dotted line at the 50th percentile of the genome length indicates the effective NG50 of the Vvill1.0 assembly. (B) The percentage of complete (CompleteSC), duplicated (CompleteDup), fragmented (Fragmented), and missing (Missing) single copy orthologous genes from Vvill1.0 and *V. sativa* identified using the BUSCO [42] software package. The eudicots_odb10 dataset (2,326 markers) was used as the library for detecting single-copy orthologs in both assemblies.

respectively, revealing a similar divergence in sequence profile in whole genome sequencing (WGS) read alignments. The *V. villosa* reads that did map to *V. sativa* had multiple single nucleotide variants and insertion–deletion mutations, suggesting that frequent small variants may also cause issues with genome alignment comparisons even though the two species belong to the same genus. The frequency of sequence variants was

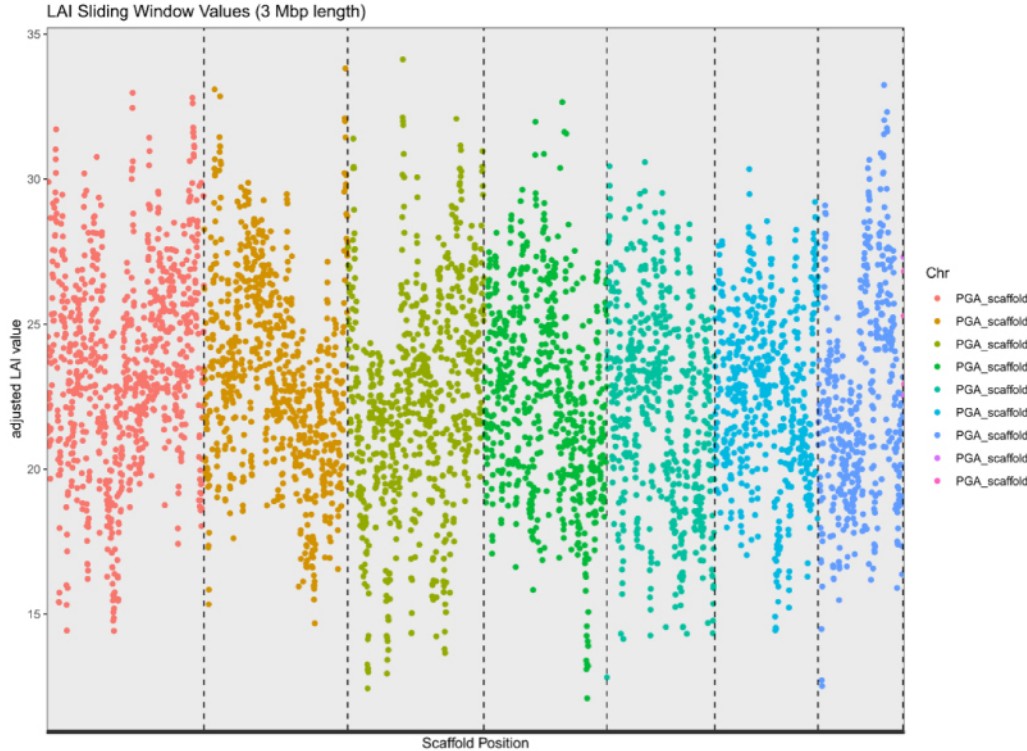

**Figure 5.** Regional differences in LAI values on the Vvill1.0 reference assembly highlighted in a sliding window analysis. Each dot is colored by the originating scaffold of the Vvill1.0 assembly and represents the LAI value in a 3 Mbp window (step = 300 kbp) of the assembly. Vertical dashed lines represent the boundaries of the major scaffolds of the assembly. Any LAI value greater than 20.0 represents the "gold" standard for assembly quality of LTR repetitive elements.

confirmed by our Freebayes (RRID:SCR_010761) analysis of short-read alignments [44]. Freebayes variant calls were used to generate a quality value (QV, or Phred [45]) score for all bases with at least 3× coverage as described previously [36]. The base QV for our Vvill1.0 assembly was 45.02, indicating a >99.99% accuracy of genome sequence compared to short-read alignments (Table 3). Read alignments of *V. villosa* short-read data to the *V. sativa* reference produced a suboptimal 14.66 QV, representing a difference in base alignment quality of three orders of magnitude compared to the Vvill1.0 assembly. Such comparative statistics do not indicate any deficiency in the *V. sativa* assembly but reflect the advantages of a species-specific reference assembly for *V. villosa* genomic analyses.

The k-mer count plot [46] for our assembly shows a prominent peak at ~35× coverage representing k-mers from heterozygous sequences, and a much smaller peak at ~70× coverage representing k-mers from homozygous sequences (Figure 6). The approximately two-fold higher count of heterozygous compared to homozygous k-mers is in agreement with the high level of heterozygosity (3.1%) estimated by GenomeScope using the *V. villosa* short-reads as input (Figure 2). This elevated heterozygosity is likely a result of the cross-pollinating nature of *V. Villosa* compared with the selfing nature of *V. sativa* [39]. We note that the "read-only" k-mer peak, representing k-mers observed in the short-reads but not in the assembly, indicates that some unique heterozygous sequence is not completely represented in Vvill1.0. This is likely a result of the removal of duplicated sequences

**Table 3.** Read mapping statistics of Vvill1.0 and *V. sativa* genome assemblies using *V. villosa* short-reads.

| Assembly quality statistics | Vvill1.0 | *V. sativa*[a] |
|---|---|---|
| Reads mapped (%) | 98.6 | 47.0 |
| Genome coverage (%) | 99.9 | 20.8 |
| Base QV | 45.0 | 14.7 |
| k-mer completeness | 81.6 | 5.6 |
| k-mer error rate | $8.1 \times 10^{-6}$ | 0.1 |
| k-mer based QV | 50.9 | 11.7 |
| SV-DEL | 27,169 | 17,808 |
| SV-DUP | 5,659 | 8,827 |
| SV-BND | 101,348 | 233,506 |
| LOW_COV_PE | 91,325 | 409,606 |
| LOW_NORM_COV_PE | 67,103 | 391,665 |
| HIGH_SPAN_PE | 1,928 | 172,241 |
| HIGH_COV_PE | 19,400 | 120,215 |
| HIGH_NORM_COV_PE | 19,899 | 88,253 |
| HIGH_OUTIE_PE | 276 | 18,762 |
| HIGH_SINGLE_PE | 79 | 204,603 |
| STRECH_PE | 23,819 | 28,103 |
| COMPR_PE | 106,336 | 178,393 |

[a]Comparisons are from *V. villosa* short-reads mapped to the *V. sativa* reference genome to demonstrate the utility of a separate reference genome for the former species. Variant calls by Freebayes [44] were used to calculate the Base QV for all bases with at least 3× coverage. K-mer completeness, k-mer error rate, and k-mer-based QV were calculated using merqury [46]. All structural variants (SV-DEL: deletions, SV-DUP: duplications, and SV-BND: trans-contig associations) were identified using LUMPY-SV [47]. Rows with a "PE" suffix indicate features identified by FRCbam [48], and the detailed definitions for each can be found in the original publication. Brief descriptions are as follows: LOW_COV_PE: regions of low read coverage; LOW_NORM_COV_PE: regions of low coverage of normal PE reads; HIGH_SPAN_PE: regions with high numbers of read pairs that map to different contigs/scaffolds; HIGH_COV_PE: regions of high read coverage; HIGH_NORM_COV_PE: regions of high coverage of normal PE reads; HIGH_OUTIE_PE: regions with high numbers of misoriented pairs; HIGH_SINGLE_PE: regions with high numbers of unmapped pairs; STRECH_PE: regions with high compression/expansion statistics; COMPR_PE: regions with low compression/expansion statistics.

resulting from the PacBio IPA assembly and the purge_dups workflow we used to generate Vvill1.0. The k-mer histogram plots are highly sensitive to the absence of single nucleotide variants that were likely present in purged duplicated regions, so their absence is less likely to impact future DNA sequence alignment surveys. This notable absence of k-mer frequency does provide a cautionary tale, as the purging of additional duplicated sequences would only exacerbate issues with genome representation, as mentioned above.

The discrepancies in alignment quality noted in our comparisons of *V. villosa* short-read data with the *V. sativa* reference assembly led us to question if there were significant structural discrepancies between the two species. The accuracy of the structural variant prediction was assessed using LUMPY-SV (RRID:SCR_003253) [47] to call structural variants and FRCbam (RRID:SCR_005189) [48] to identify features or suspicious regions of the assembly based on read alignments, with *V. villosa* short-reads as input. The short-read alignments to the *V. sativa* genome assembly predicted 260,141 structural variants, with the majority predicted as complex structural variants (233,506). This is nearly twice the number of structural variants predicted compared to aligning the same sequence reads to the *V. villosa* assembly (134,176). Further, the short-read alignments to the *V. sativa* genome had a substantially higher count of discordant genomic features than alignments to our *V. villosa* assembly (Table 3). These results suggest that smaller-scale (50–50,000 bp) structural variations in genome sequence exist between the two species.

Larger changes in genome structure were classified by identifying any candidate syntenic regions through whole-genome alignment. Minimap2 (RRID:SCR_018550) was used



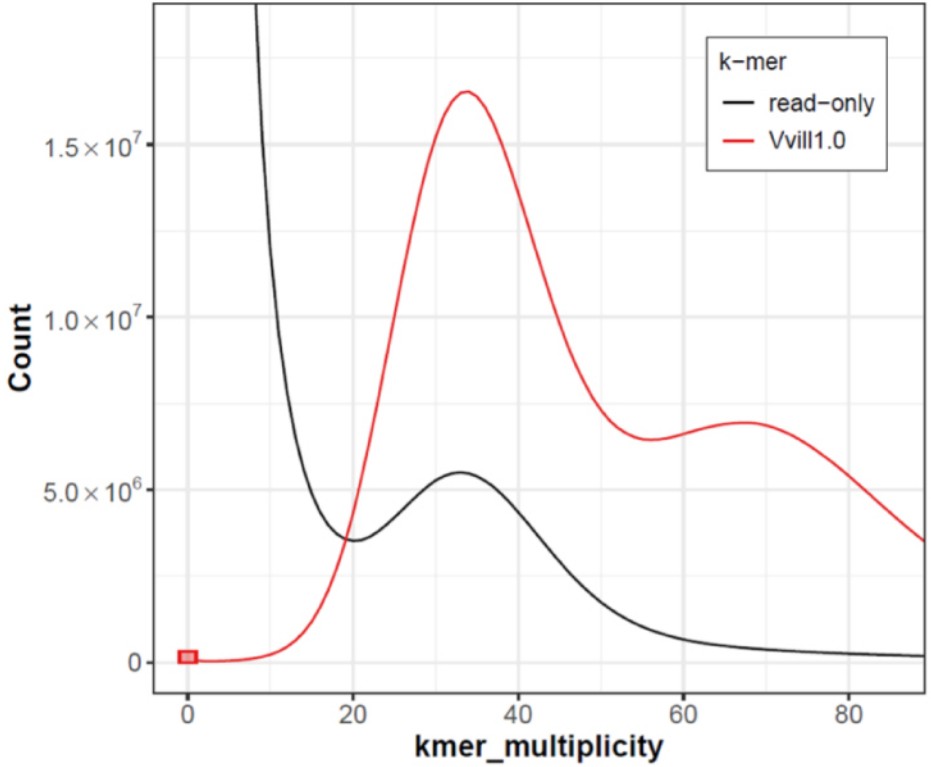

**Figure 6.** K-mer assembly spectra plot generated by merqury [46] showing the distribution of k-mers ($k$ = 21) found in the Illumina short-read set (black, read-only) and k-mers found in our Vvill1.0 assembly (red, Vvill1.0). The red bar at zero multiplicity indicates k-mers found only in the assembly. The read-only peak at ~35× likely represents heterozygous variants missing from the assembly.

to identify pairwise alignments between our Vvill1.0 assembly and the *V. sativa* assembly using an alignment cutoff of 100,000 bp segments or greater [49]. The results were displayed as a circos plot (Figure 7A) [50]. Some conserved segments of chromosomes were observed, but most alignments are spread out between the chromosome scaffolds of the two species. This variation in the genomic architecture suggests relaxed constraints on gene organization across these closely related species. By contrast, a similar whole genome alignment of the reference genomes of two other legume species shows better conservation of syntenic regions (Figure 7B). The chromosomal reorganization between these two species may underlie some of the phenotypic variations between them and further highlights the importance of having a species-specific genome reference assembly for future studies of wild and cultivated vetch species.

## Genome annotation

Classification of all genic content and repetitive loci within Vvill1.0 was performed to increase its utility as a genomic resource. A list of canonical *V. villosa* repetitive elements was generated *de novo* using the EDTA version 2.0.0 software tool (RRID:SCR_022063) [51] with the "sensitive" setting to enable RepeatModeler (RRID:SCR_015027) recovery of transposable elements. The set of *V. villosa* canonical repetitive elements was then used as a custom library input to RepeatMasker version 4.0.6 (RRID:SCR_012954) [52], which was in

**A**

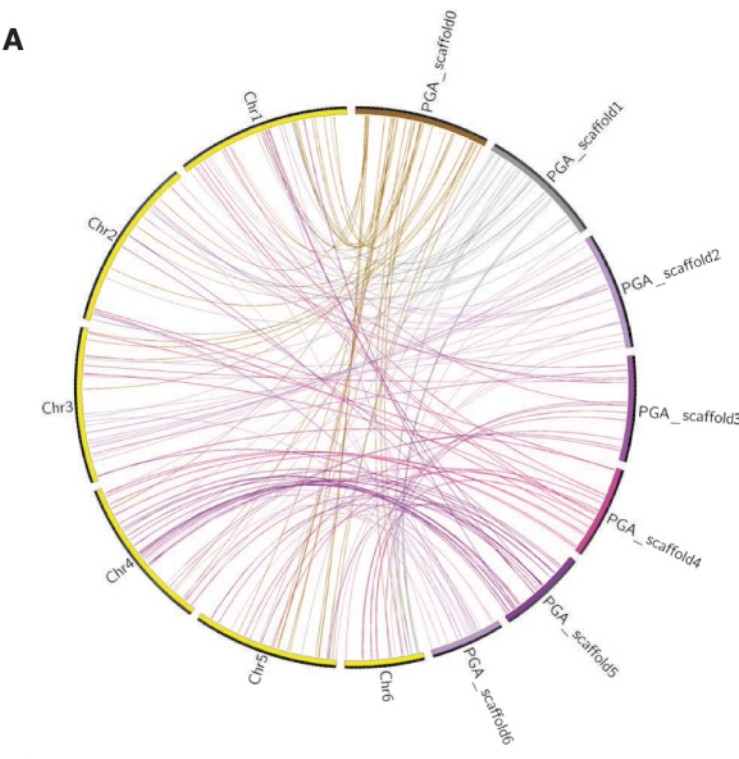

**B**

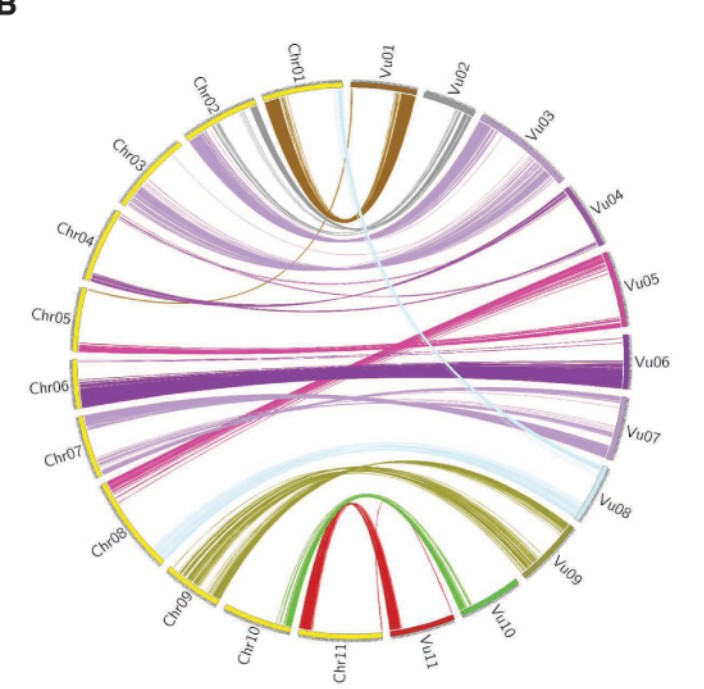

**Figure 7.** Circos plot showing syntenic regions shared between (A) the *V. sativa* assembly (yellow outer bands) and Vvill1.0 (multi-colored outer bands) genomes, or (B) the *Phaseolus vulgaris* (yellow) and *vigna unguiculata* (multi-colored) genomes [48]. Ribbons (colored matching the Vvill1.0 scaffolds (A) or the *vigna unguiculata* chromosomes (B)) represent the pairwise alignments of 100 kbp or larger identified using minimap2 [49].

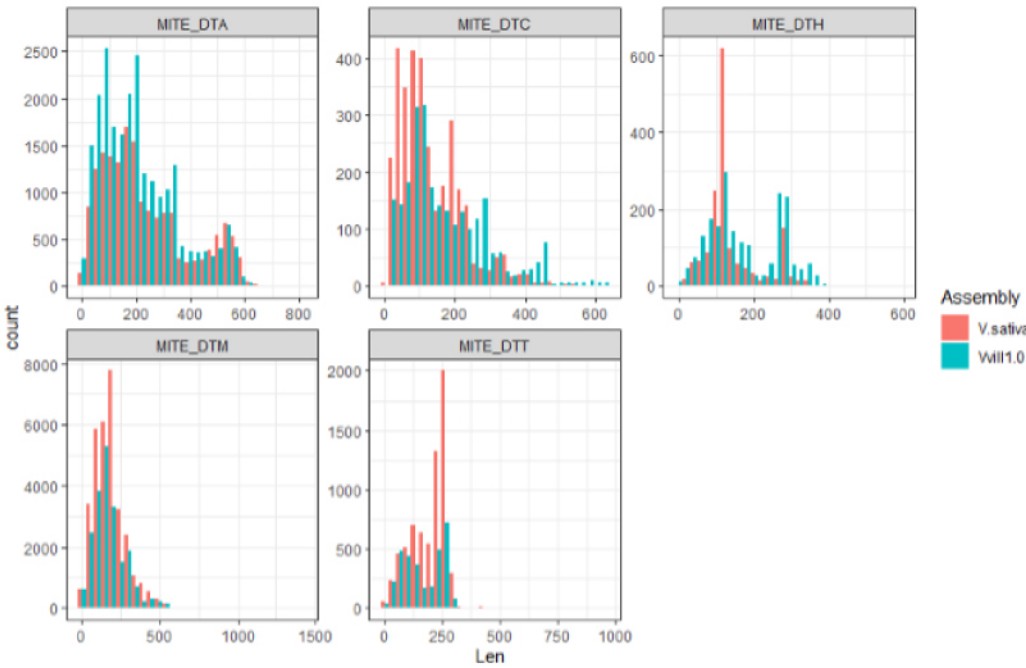

**Figure 8.** Length distribution of MITE repeats in *V. sativa* and *V. villosa.* MITE families are indicated by a suffix after the underscore in each subplot's title, and follow the Repbase (https://www.girinst.org/repbase/) naming classifications.

**Table 4.** Repetitive element content of *V. villosa*.

| Repetitive elements | | Number | Cumulative length (bp) | Percentage of genome |
|---|---|---|---|---|
| Retroelements[a] | | 1,080,921 | 830,932,491 | 60.0 |
| | LINEs | 2,982 | 1,105,274 | 0.1 |
| | LTRs | 1,077,939 | 829,827,217 | 59.9 |
| DNA transposons | | 802,725 | 224,578,692 | 16.2 |
| Unclassified | | 221,628 | 53,995,075 | 3.9 |
| Simple repeats | | 193,714 | 11,729,117 | 0.9 |
| Low complexity | | 30,795 | 1,617,938 | 0.1 |
| **Total** | | **2,329,783** | **1,122,853,313** | **81.1** |

[a]Long interspersed nuclear elements (LINE), long terminal repeats (LTR).

turn used to soft-mask the Vvill1.0 assembly. The repetitive content was similar to the *V. sativa* reference assembly, with 81.1% of the assembly consisting of identified repeats in Vvill1.0 (Table 4), compared to the 83.9% repetitive content in *V. sativa*. Comparisons of repetitive element lengths revealed few discrepancies in repeat content between the two vetch assemblies with similar distributions of repeat fragment sizes for nearly all classes. A notable discrepancy was identified in the size distributions of miniature inverted-repeat transposable elements (MITE), where larger MITE_DTH and MITE_DTC elements were more prevalent in *V. villosa* and larger MITE_DTT elements were more prevalent in *V. sativa* (Figure 8). This suggests that differential expansion and amplification bursts of MITEs may have occurred in both lineages after their divergence.

All coding sequences in the Vvill1.0 assembly were annotated using a combination of *ab initio* prediction and RNAseq evidence. RNAseq reads from Ali *et al.* (2023) [53] were



**Table 5.** Gene annotation summary statistics.

| Features | Vvill1.0 | *V. sativa*[a] |
|---|---|---|
| Protein-coding genes | 53,321 | 53,218 |
| Average exons per gene | 4.6 | 4.4 |
| Average exon length (bp) | 207.4 | 223.4 |
| Average intron length (bp) | 434.0 | 415.1 |

[a]Summary statistics for the *V. sativa* assembly were taken from [19].

**Table 6.** Number of genes with functional annotations identified using different databases.

| Database | | Number annotated | Percent annotated |
|---|---|---|---|
| NCBI-NR | | 43,455 | 81.5 |
| UniProt | | 32,445 | 60.9 |
| EggNOG | Pfam | 37,949 | 71.2 |
| | KEGG_pathway | 12,887 | 24.2 |
| | KEGG_KO | 20,055 | 37.6 |
| | GO | 20,786 | 39.0 |
| **Total annotated** | | 43,626 | 81.8 |
| **Total** | | 53,312 | |

aligned to the soft-masked Vvill1.0 assembly using the STAR alignment tool version 2.7.9 (RRID:SCR_004463) with the "genomeGenerate" runtime mode. Gene prediction was performed using BRAKER2 (v2.1.6; RRID:SCR_018964) [54] with the soft-masked version of the Vvill1.0 assembly mentioned above as the template. We identified 53,321 protein-coding genes (Table 5), which was nearly equivalent to the number of protein-coding genes (53,218) annotated in the *V. sativa* reference assembly.

Putative functions of identified coding sequences were identified through the alignment of predicted protein amino acid sequences of *V. villosa* genes against the UniProt database (release 2022_02) and the National Center for Biotechnology Information (NCBI) non-redundant database using the DIAMOND alignment tool version 2.0.14.152 (RRID:SCR_016071) [55]. The top scoring hit was chosen for each sequence (see GigaDB supplementary data files uniport_anno.tsv and ncbi-nr_anno.tsv for the DIAMOND output data for the UniProt and NCBI non-redundant databases, respectively) [56]. Protein sequences were also aligned against the EggNOG database version 5.0.2 using EggNOG-mapper version 2.1.8 (RRID:SCR_021165) in order to assign Kyoto Encyclopedia of Genes and Genomes (KEGG) pathways and KEGG orthologous groups to each sequence [57] (see GigaDB supplementary data file eggnog.tsv for the output data from EggNOG-mapper). The outcome was the annotation of 43,626 (81.8%) predicted protein-coding genes with at least one function (Table 6).

## Phylogenetic tree construction

Large structural variations identified from chromosome scaffolds of *V. sativa* and *V. villosa* led us to explore the significant divergence in the genic sequence of these two species. Using a similar strategy to Xi *et al.* [13], we used the protein-coding sequence of nine legume species (Table 7) to estimate gene orthogroups. OrthoFinder version 2.5.4 (RRID:SCR_017118) was used to cluster all annotated genes into orthogroups with default parameters [58]. Orthogroup gene assignments were compared across species using the UpSetR package version 1.4.0 [59] in R 4.2.1. Newick files generated by Orthofinder were visualized in the etetoolkit's "treeview" utility (RRID:SCR_016916) (Figure 9). The Vvill1.0

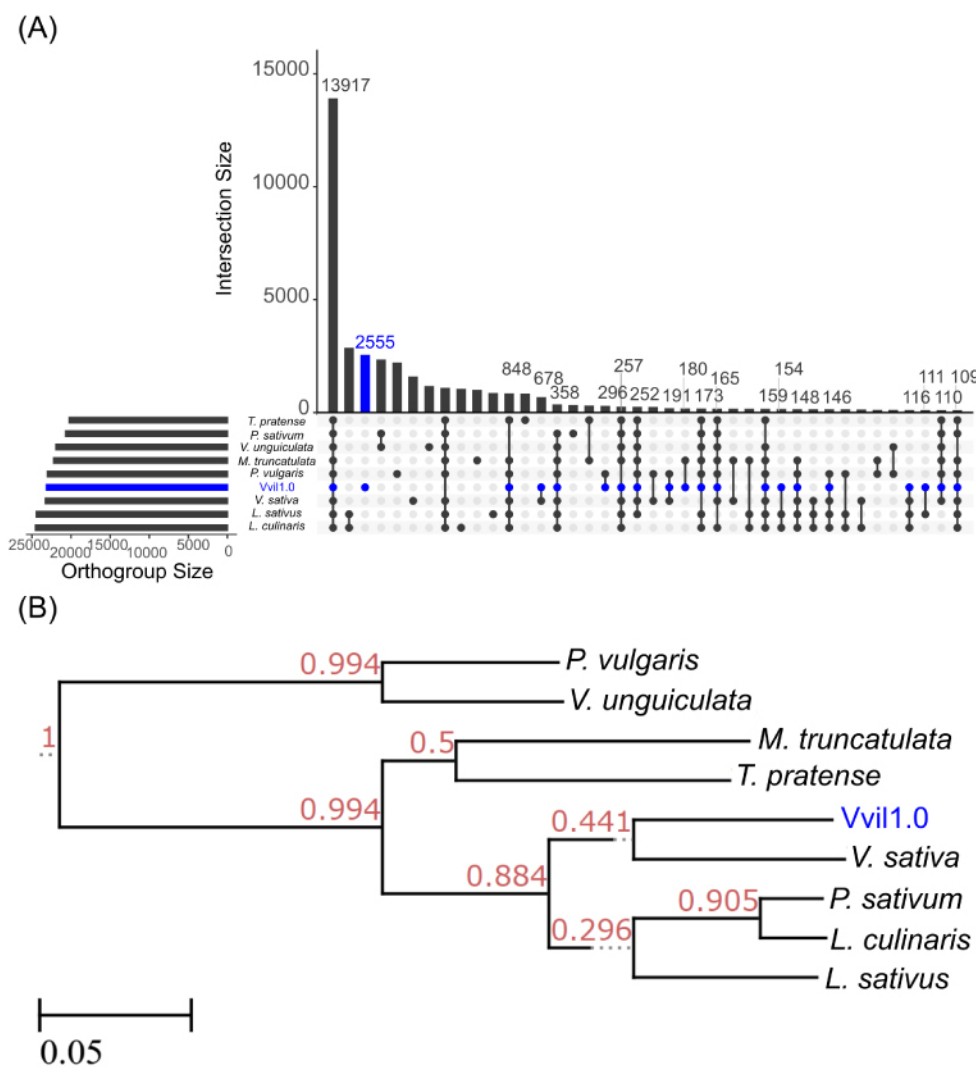

**Figure 9.** Orthogroup gene comparisons among nine legume species. An upset plot of identified orthogroups (A) suggests that *V. villosa* (blue) has the most unique annotated orthogroups of all compared legume species. Orthogroup dendrogram (B) showing the ortholog-derived relationship of *V. villosa* to other legume species. Values at each node indicate the bootstrap support for each node based on the magnitude of relative error (MRE) test that is the default in the Orthofinder software tool.

assembly was found to have the most exclusive orthogroups at 2,555 total orthogroups (Figure 9A). Gene orthogroup dendrograms (Figure 9B) suggest that the gene orthogroup content is similar between the *V. sativa* and Vvill1.0 reference assemblies despite the previously mentioned differences between the two assemblies (Figure 7). We note that this dendrogram does not match the organization of the Fabeae tribe members proposed by Macas *et al.* [24]. This is mostly due to differences in comparisons between genetic features: where Macas *et al.* [24] compared repetitive-element conservation, our study compared gene-orthogroup sequence conservation. Repetitive elements are often not under selective pressures and are more frequently subject to mutation [60, 61]. This fact makes them more informative in comparisons of closely related members of the same species. Comparison of conserved gene orthogroups can accurately reveal the divergent lineages of different

**Table 7.** List of the species and their associated genome assemblies used in this study.

| Species | Source of data | Version |
|---|---|---|
| *Vicia villosa* | This project | 1.0 |
| *Vicia sativa* | GigaDB | 1.0 |
| *Vigna unguiculata* | Phytozome | 1.0 |
| *Phaseolus vulgaris* | Phytozome | 2.0 |
| *Lathyrus sativus* | Phytozome | 1.0 |
| *Lens culinaris* | Phytozome | 2.0 |
| *Medicago truncatula* | INRA | MtA17 r5 |
| *Pisum sativum* | URGI | 1a |
| *Trifolium pratense* | GenBank | 1.1 |

species; however, such comparisons are only possible after constructing representative genome assemblies. Our assembly of the Vvill1.0 reference genome finally allows the accurate placement of *V. villosa* within the Fabeae tribe using conserved gene sequence analysis.

## REUSE POTENTIAL

Our chromosome-scale genome assembly of *V. villosa* provides the foundation for a genetic improvement program for an important cover crop and forage species. Beyond its practical uses, the assembly shows a substantial difference in genome structure compared to a recently released member of the same genus, *V. sativa*. These structural differences are in contrast to the conservation of gene orthologs shared by the two species, which suggests that the *V. villosa* assembly may provide an interesting outgroup in comparisons of leguminous plant genomes. Finally, the documentation of the methods used to resolve a highly heterozygous genome assembly will be useful in resolving issues with the assemblies of other outcrossing plant species. Specifically, to our knowledge, we are the first to document telomeric "bouquet" patterns during scaffolding using chromatin capture. Hence, these methods and our resulting genome assembly will be useful to a wider group of researchers interested in assembling genomes from leguminous plant species.

## AVAILABILITY OF SOURCE CODE AND REQUIREMENTS

The Themis-ASM assembly validation workflow is available at the following GitHub repository: https://github.com/tdfuller54/Themis-ASM. All other custom scripts used to process the data and generate the figures can be found at the following GitHub repository: https://github.com/njdbickhart/ForageAssemblyScripts.

## DATA AVAILABILITY

All raw sequence data used in the genome assembly and validation can be found in the NCBI's Sequence Read Archive under the Bioproject accession PRJNA868110. The genome accession for the Vvill1.0 assembly is under the NCBI accession JAROZA000000000. The transcript data used for annotation [53] is under the NCBI Bioproject accession PRJNA833581. Other data are available via GigaDB [56].

## ABBREVIATIONS

KEGG, Kyoto Encyclopedia of Genes and Genomes; LAI, LTR assembly index; LINE, long interspersed nuclear elements; LTR, long terminal repeat; MITE, miniature inverted-repeat transposable elements; NCBI, National Center for Biotechnology Information; PE,

paired-end; QV, quality value; SMRT, single-molecule real-time sequencing; SNP, single nucleotide polymorphism; WGS, whole genome sequencing.

## DECLARATIONS

### Ethics approval and consent to participate

The authors declare that ethical approval was not required for this type of research.

### Competing interests

HM is an employee of Phase Genomics (Seattle, WA, USA). DMB is an employee of Hendrix-Genetics (Boxmeer, the Netherlands). MJM is an employee of Bayer Crop Science (Chesterfield, MO, USA). All other authors declare that they have no competing interests.

### Authors' contributions

LMK, TPLS, and MLS generated the genome WGS and Omni-C data. SA generated the transcript sequence data. DMB and TPLS assembled the genome, and DMB purged the duplicates. MJM secured the resources for tissue propagation and secured the Hi-C genome sequences. TH propagated the tissue of the HV-30 line for sequencing. HM generated the scaffolds from the Hi-C read alignments. DMB and TF ran the analysis of the assembly. All authors read and contributed to the final version of the manuscript.

### Funding

USDA, Agricultural Research Service, 5090-31000-026-00D, DMB; USDA, Agricultural Research Service, 5090-21000-071-00D, MLS; USDA, Agricultural Research Service, 5090-21000-001-00D, HR; USDA, Agricultural Research Service, 3040-31000-100-00D, TPLS; USDA, National Institute of Food and Agriculture, 2018-67013-27570, HR; USDA, National Institute of Food and Agriculture, 2018-67013-27570, LKK.

### Acknowledgements

We thank Dr. Kristen Kuhn, Kelsey McClure, and Dr. Jennifer McClure for technical assistance. This project was supported in part by an appointment (of SA) to the Research Participation Program at the US Dairy Forage Research Center, ARS-USDA, administered by the Oak Ridge Institute for Science and Education through an interagency agreement between the U.S. Department of Energy and ARS-USDA. ORISE is managed by ORAU under DOE contract number DE-SC0014664. All opinions expressed in this paper are the author's and do not necessarily reflect the policies and views of USDA, DOE, or ORAU/ORISE. Sequencing and resources for this project were provided by the Noble Research Institute. The USDA does not endorse any products or services. Mentioning of trade names is for information purposes only. The USDA is an equal opportunity employer.

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
