## [Editor Report]

Editor’s AssessmentThe hairy vetch Vicia villosa is an annual legume widely used as a cover crop due to its ability to withstand harsh winters. Here a new a 2.03GB reference-quality genome is presented, assembled from PacBio HiFi long-sequence reads and Hi-C scaffolding. After adding some more methodological details and long-terminal repeat (LTR) assembly index (LAI) analysis the assembly quality and metrics look quite convincing as a chromosome-scale assembly. This resource hopefully providing the foundation for a genetic improvement program for this important cover crop and forage species.

---

## [Reviewer Report]

Upload additional filesDRR-202303-02/form/Reviewer comments.docxReviewer name and names of any other individual's who aided in reviewer Rong LiuDo you understand and agree to our policy of having open and named reviews, and having your review included with the published papers. (If no, please inform the editor that you cannot review this manuscript.)YesIs the language of sufficient quality?YesPlease add additional comments on language quality to clarify if needed
Are all data available and do they match the descriptions in the paper? YesAdditional CommentsAre the data and metadata consistent with relevant minimum information or reporting standards? See GigaDB checklists for examples <a href="http://gigadb.org/site/guide" target="_blank">http://gigadb.org/site/guide</a>YesAdditional CommentsIs the data acquisition clear, complete and methodologically sound?YesAdditional CommentsIs there sufficient detail in the methods and data-processing steps to allow reproduction?YesAdditional CommentsIs there sufficient data validation and statistical analyses of data quality? NoAdditional CommentsIs the validation suitable for this type of data?YesAdditional CommentsIs there sufficient information for others to reuse this dataset or integrate it with other data?YesAdditional CommentsAny Additional Overall Comments to the AuthorRecommendationMinor Revision

---

## [Reviewer Report]

Reviewer name and names of any other individual's who aided in reviewer haifei huDo you understand and agree to our policy of having open and named reviews, and having your review included with the published papers. (If no, please inform the editor that you cannot review this manuscript.)YesIs the language of sufficient quality?YesPlease add additional comments on language quality to clarify if needed
Are all data available and do they match the descriptions in the paper? NoAdditional CommentsAre the data and metadata consistent with relevant minimum information or reporting standards? See GigaDB checklists for examples <a href="http://gigadb.org/site/guide" target="_blank">http://gigadb.org/site/guide</a>YesAdditional CommentsIs the data acquisition clear, complete and methodologically sound?YesAdditional CommentsIs there sufficient detail in the methods and data-processing steps to allow reproduction?YesAdditional CommentsIs there sufficient data validation and statistical analyses of data quality? NoAdditional CommentsIs the validation suitable for this type of data?YesAdditional CommentsIs there sufficient information for others to reuse this dataset or integrate it with other data?YesAdditional CommentsAny Additional Overall Comments to the AuthorFuller et al. conducted an interesting work on the Vicia villosa genome study, which could be beneficial for the science community. However, there are some concerns about this work before it can be published.    1. Introduction  The MS seems to indicate the V.villosa genome is important for breeding, and it is an ideal legume that can grow in winter. But the coming analysis and results are missing to address this. The authors should include additional analysis, at least in the gene annotation session, to indicate what genes are potentially associated with the improvement of genetic-based selection and the ability to grow in winter conditions.  After reading the MS, it looks like it mainly focuses on the comparison of the V.vilsoa genome and the V.sativa genome. Please indicate why it is important to do so and provide more background on V.sativa in the introduction.  Line 59. It is too sudden to start to describe high heterozygosity as still in the challenge without directly linking to V.villosa. The authors need to include the background that V.villosa is heterozygous first, then talk about how challenging it is to generate an assembly.   2. Methods Line 112: Why is the estimation based on K-mer size quite different from the generated assembly size? The authors’ explanation is weak and needs an in-depth and better explanation of these unexpected results. Did you see any similar observations in other studies? Please give examples(citations). Line 121: Any reason not to use the commonly used HiFi assembler HFi-asm? Line 142-143: Did you have a file to record which genome regions you have introduced the breaks and how this step was performed? Line 158: the unit bp changed into Mb for better comparison Line 160: Here, you should use contig N50 rather than scaffold N50 to indicate the quality of the gnome. And you need to compare the contig N50 with the V.sativa.   3. DATA VALIDATION AND QUALITY CONTROL Should perform BUSCO and LAI to assess the quality of the genome in the main text.   4 Phylogenetic tree construction Soybean is an important legume species, and it will make this result more useful and interesting for readers. You should include the Wm82 V4 genome for this analysis. And the version of other legume species’ genomes needs to be indicated.  5 Figures Figure 3 HiC alignment map shows near 600Mb genomes can not be scaffolded into a genome. Any reason? What is the green dot point in the figure?  Figure 4 b, the BUSCO of Vvil1.0 is much higher than V.stativa. Any reason? And no description of how you perform the BUSCO analysis in the main text.  Figure 6 Circle plot, would that possible to rename the scaffold as a chromosome based on the alignment between V.sativa and V.vil? 
RecommendationMajor Revision